# Potential for Drug Repositioning of Midazolam for Dentin Regeneration

**DOI:** 10.3390/ijms20030670

**Published:** 2019-02-04

**Authors:** Takeo Karakida, Kazuo Onuma, Mari M. Saito, Ryuji Yamamoto, Toshie Chiba, Risako Chiba, Yukihiko Hidaka, Keiko Fujii-Abe, Hiroshi Kawahara, Yasuo Yamakoshi

**Affiliations:** 1Department of Biochemistry and Molecular Biology, School of Dental Medicine, Tsurumi University, 2-1-3 Tsurumi, Tsurumi-ku, Yokohama 230-8501, Japan; karakida-t@tsurumi-u.ac.jp (T.K.); saito-mari@tsurumi-u.ac.jp (M.M.S.); yamamoto-rj@tsurumi-u.ac.jp (R.Y.); chiba-r@tsurumi-u.ac.jp (R.C.); 2National Institute of Advanced Industrial Science & Technology, Central 6, 1-1-1 Higashi, Tsukuba, Ibaraki 305-8566, Japan; k.onuma@aist.go.jp; 3Research Center of Electron Microscopy, School of Dental Medicine, Tsurumi University, 2-1-3 Tsurumi, Tsurumi-ku, Yokohama 230-8501, Japan; chiba-t@tsurumi-u.ac.jp; 4Department of Dental Anesthesiology, School of Dental Medicine, Tsurumi University, 2-1-3 Tsurumi, Tsurumi-ku, Yokohama 230-8501, Japan; 2911002@stu.tsurumi-u.ac.jp (Y.H.); fujii-keiko@tsurumi-u.ac.jp (K.F.-A.); kawahara-h@tsurumi-u.ac.jp (H.K.)

**Keywords:** drug repositioning, cell, dentin, hydroxyapatite, nanorod, nanoparticle

## Abstract

Drug repositioning promises the advantages of reducing costs and expediting approval schedules. An induction of the anesthetic and sedative drug; midazolam (MDZ), regulates inhibitory neurotransmitters in the vertebrate nervous system. In this study we show the potential for drug repositioning of MDZ for dentin regeneration. A porcine dental pulp-derived cell line (PPU-7) that we established was cultured in MDZ-only, the combination of MDZ with bone morphogenetic protein 2, and the combination of MDZ with transforming growth factor-beta 1. The differentiation of PPU-7 into odontoblasts was investigated at the cell biological and genetic level. Mineralized nodules formed in PPU-7 were characterized at the protein and crystal engineering levels. The MDZ-only treatment enhanced the alkaline phosphatase activity and mRNA levels of odontoblast differentiation marker genes, and precipitated nodule formation containing a dentin-specific protein (dentin phosphoprotein). The nodules consisted of randomly oriented hydroxyapatite nanorods and nanoparticles. The morphology, orientation, and chemical composition of the hydroxyapatite crystals were similar to those of hydroxyapatite that had transformed from amorphous calcium phosphate nanoparticles, as well as the hydroxyapatite in human molar dentin. Our investigation showed that a combination of MDZ and PPU-7 cells possesses high potential of drug repositioning for dentin regeneration.

## 1. Introduction

Tissue engineering is a multidisciplinary science. Applications of tissue engineering are founded on three components; a cell source, a scaffold, and bioactive molecules. In the field of dental tissue engineering, various soft and hard dental tissues have been regenerated in vitro using stem cells [1]. Dental pulp stem cells (DPSCs) have been isolated with various techniques and used for studies related to the cell differentiation potential and scaffolding for tissue regeneration [2,3]. DPSCs can differentiate into multiple cell types, including odontogenic and osteogenic cells, osteocytes, chondrocytes, vascular cells, neurons, and hepatocytes. In addition, DPSCs are used to generate induced pluripotent stem (iPS) cells. 

In the dental research field, dental pulp-derived cells that express alkaline phosphatase (ALP) and odontoblastic marker genes, and that form precipitated nodules when cultured in a medium containing β-glycerophosphate and ascorbic acid, are generally assumed to be differentiated odontoblast-like cells. A number of odontoblastic cell lines have been established from rat [4,5], mouse [6], cattle [7], pig [8], and human [9,10] dental pulp tissues. Odontoblastic differentiation is induced by many factors, which are associated with ectoderm–mesenchyme molecular interactions. Various factors participate in the regulation of dental pulp cell differentiation, such as bone morphogenetic protein (BMP) [11,12], fibroblast growth factors (FGF) [13], and transforming growth factor beta (TGF-β) [14,15]. 

Drug repositioning is when new therapeutic applications are identified for existing drugs. In addition to the studies on dental pulp cells expressing cytokines, such as BMP, FGF, and TGF-β, existing drugs for treating Alzheimer’s disease have been reported to promote dentin regeneration [16]. In Japan, Alzheimer’s drugs such as donepezil hydrochloride (Aricept) have been widely used as acetylcholinesterase inhibitors that prevents acetylcholine, an excitatory neurotransmitter in the vertebrate nervous system. This study focuses on repositioning midazolam (MDZ), which controls gamma-aminobutyric acid (GABA), the principal inhibitory neurotransmitter in the mammalian central nervous system. Midazolam (MDZ) is a chemically synthesized imidazobenzodiazepine derivative that possesses pharmacological effects as a hypnotic, sedative, and anesthetic, has anti-anxiety and anticonvulsant properties, and acts as muscle relaxant [17]. In the dental field, MDZ has been used mainly as a sedative prior to dental anesthesia. The intravenous MDZ formulation has been recommended as a first-line drug for treating convulsive status epilepticus [18]. Intravenous MDZ has not been approved to treat status epilepticus in most countries, but it has been used off-label for patients in Japan. In tumor and cancer cells, MDZ induces cellular apoptosis by regulating the caspase pathways, endoplasmic reticulum stress, autophagy, and the cell cycle [19,20,21,22]. Few studies have focused on the effect of MDZ on dental pulp cells, although an in vitro study using human mesenchymal stem cells (hMSCs) has shown that MDZ inhibits ALP activity and calcium deposition in hMSCs, suggesting a suppressive effect on osteogenic differentiation [23]. In this study, using dental pulp cells, we examine the potential for drug repositioning of MDZ for dentin regeneration at the cell biological, genetic, protein, and crystal engineering levels. 

## 2. Results

### 2.1. Combined Effect of MDZ with BMP2 or TGF-β1 on Differentiation of the PPU-7 Cell Line

MDZ is a short-acting benzodiazepine derivative with an imidazole structure and a molecular weight of 325.77 g/mol (Figure 1a). To investigate the effect of MDZ on dentin regeneration, we used a porcine animal model to establish dental pulp-derived cell lines (PPU) and ultimately selected the PPU-7 cell line for the present study (see Appendix A). Because alkaline phosphatase (ALP) is considered the initial marker for the differentiation of mesenchymal cells into hard tissue-forming cells such as osteoblasts or odontoblasts [24,25], we investigated the effects of MDZ on ALP activity in the PPU-7 cell line by using a quantitative colorimetric method with a p-nitrophenylphosphate as the substrate and examined the combination of MDZ (0-10 μM) with a recombinant human BMP2 (rhBMP2) (500 ng/mL) or a recombinant human TGF-β1 (rhTGF-β1) (1 ng/mL) (Figure 1b). When the ALP activity level of the control (i.e., 0 μM MDZ, 0 ng/mL rhBMP2, 0 ng/mL rhTGF-β1) was set at 1.0, the addition of MDZ-only significantly enhanced ALP activity in PPU-7 cells in a concentration-dependent manner, especially the ALP activity at 10 μM MDZ (i.e., 10 μM MDZ, 0 ng/mL rhBMP2, 0 ng/mL rhTGF-β1), which was 1.75-fold higher than the control. BMP2-only (i.e., 0 μM MDZ, 500 ng/mL rhBMP2, 0 ng/mL rhTGF-β1) and TGF-β1-only (i.e., 0 μM MDZ, 0 ng/mL rhBMP2, 1 ng/mL rhTGF-β1) also slightly or significantly enhanced the ALP activity. The combination of rhBMP2 or rhTGF-β1 with MDZ (MDZ and BMP2 or MDZ and TGF-β1) (5 or 10 μM) slightly increased ALP activity (1.14–1.27-fold for MDZ and BMP2, and 1.28–1.29-fold for MDZ and TGF-β1) compared to the control. ALP staining for the mineral-induced PPU-7 cells displayed blue colored staining images (Figure 1c). The cells were densely distributed on the plate of MDZ-only, whereas the combinations of MDZ and BMP2 as well as MDZ and TGF-β1 displayed a low density.

### 2.2. Effect of MDZ on Temporal Changes in Gene Expression of PPU-7 Cell Line

Since the MDZ-only treatment was more effective in enhancing ALP activity and inducing mineralization in PPU-7 cells than the combination treatment of MDZ with BMP2 or TGF-β1, we investigated the effect of MDZ-only on gene expression in the PPU-7 cell line. The gene expression of a panel of odontoblastic, osteoblastic and chondrocytic markers in PPU-7 cells at one and seven days after MDZ treatment was analyzed by quantitative polymerase chain reaction (qPCR) (Figure 2). For odontoblastic markers (Figure 2a), we quantified the mRNA expression levels of matrix metalloprotease 2 (*MMP2*) and two products from the full-length *DSPP* transcript: a segment containing the dentin glycoprotein and dentin phosphoprotein (DGP and DPP) coding region (*DSPP-v1*) and a smaller segment specific for the dentin sialoprotein (DSP)-only transcript (*DSPP-v2*). At day one, the expression levels of *MMP2*, *DSPP-v1*, and *DSPP-v2* in cells cultured with MDZ were not significantly different from those in cells cultured without MDZ. In contrast, at day seven, the three mRNA expression levels in cells cultured with MDZ were significantly higher (3.8-fold for *MMP2*, 1.67-fold for *DSPP-v1*, and 2.2-fold for *DSPP-v2*) than those in cells cultured without MDZ. We also amplified osteocalcin (OC) and runt-related transcription factor 2 (*RUNX2*) as osteoblastic markers (Figure 2b) and type II collagen (Col II) and aggrecan (*ACAN*) as chondrocytic markers (Figure 2c). There were no significant differences in the four mRNA expression levels between cells cultured with and without MDZ at either day one or day seven. We interpret these findings as evidence that MDZ is required for the differentiation of PPU-7 cells into odontoblasts.

### 2.3. Effect of MDZ on Mineralization Induction of the PPU-7 Cell Line 

To obtain additional information about the effect of MDZ on mineralization inducibility, we cultured PPU-7 cells in a mineralization-inducing culture medium (Figure 3). The nodule formation and mineralization capacities of the cells were assessed with both Alizarin Red S and von Kossa staining (Figure 3a). At seven days following mineralization induction, in contrast to that of the cells not subjected to mineralization induction, the plate of the cells cultured in mineralization-inducing culture medium clearly displayed precipitated nodules by both staining methods, regardless of the addition of MDZ. 

We also quantitatively analyzed the calcium content in PPU-7 cells (Figure 3b). At five days following mineralization induction, relative to the control cells without MDZ (i.e., no MDZ), the cells administered with MDZ-only displayed a dramatically increased amount of calcium deposition (approximately 2.0-fold). 

### 2.4. Detection of DPP in Precipitated Nodules from PPU-7 Cells

We attempted to detect a dentin-specific protein, dentin phosphoprotein (DPP), in precipitated nodules induced by MDZ treatment in the PPU-7 cell line at the protein level. The precipitated nodules in PPU-7 cells were subjected to a series of three extractions: With a tris-guanidine buffer (G1), with hydrochloric acid–formic acid solution (HF), and with tris-guanidine buffer again (G2). These extractions yielded three fractions, the G1, HF, and G2 extracts, which were analyzed by SDS-polyacrylamide gel electrophoresis (SDS-PAGE) stained with SimplyBlue SafeStain (Figure 4a) and Stains-all stain (Figure 4b). In the G2 fraction obtained from PPU-7 cells incubated in mineralization-inducing culture medium, Stains-all-positive DPP doublet bands were observed at approximately 100 kDa by SDS-PAGE (Figure 4b, lanes 8 and 9). The addition of MDZ increased the intensity of the DPP doublets bands (Figure 4b, lane 8), whereas no DPP bands were detected in the extracts from PPU-7 cells cultured with the standard medium (Figure 4b, lane 7). 

### 2.5. X-ray Diffraction (XRD) Patterns of Precipitated Nodules 

Based on all the above results, we further characterized the precipitated nodules from PPU-7 cells at the crystal engineering level. Figure 5a shows microbeam XRD patterns for samples grown with (magenta curve) and without (blue curve) MDZ in the culture solution. The pattern for a cell sheet (green curve) is shown only for reference. The cell sheet exhibited a broad intense peak at approximately 2θ = 10°, which tailed to higher 2θ values. Peaks attributed to the precipitates appeared at 2θ = 26.0, 28.7, and 32.0° (black circles); they were independent of the presence or absence of MDZ. The entire XRD pattern of the precipitates from the solution with MDZ was the same as that of the precipitates from the solution without MDZ, signifying that MDZ did not affect the kind of phase precipitated. The peak widths in the precipitate patterns were broad, indicating low crystallinity and/or small crystallites. Figure 5b–e shows ideal XRD patterns for dicalcium phosphate dihydrate (CaHPO_4_·2H_2_O; DCPD) (b), octacalcium phosphate (Ca_8_(HPO_4_)_2_(PO_4_)_4_·5H_2_O; OCP) (c), β-tricalcium phosphate (β-Ca_3_(PO_4_)_2_; β-TCP) (d), and hydroxyapatite (Ca_10_(PO_4_)_6_(OH)_2_; HAP) (e), which were derived using the 2θ versus diffraction–intensity relationships in the corresponding Joint Committee on Powder Diffraction Standards (JCPDS) cards. The asterisks indicate the particular peaks used to identify each calcium phosphate phase. The XRD patterns of the precipitates did not show these particular peaks; however, the profile from 2θ = 25 to 35° resembled those for OCP and HAP. 

### 2.6. Scanning Electron Microscopy (SEM) Observations of Precipitated Nodules

A macroscopic SEM image of a cross-sectional sample (Figure 6a) showed that the morphology of the precipitates was not uniform. Both irregularly (white arrow) and regularly shaped precipitates (i.e., μm-scale precipitates with a ball-like appearance, yellow arrows) were observed. A magnified image of a region with these ball-like precipitates (Figure 6b) showed that they consisted of aggregated nanoparticles. The size of each nanoparticle was approximately 50 nm or less. The isotropic morphology of the elementary particles made it difficult to ascertain the phase of the calcium phosphate, so the precipitates were observed using TEM.

### 2.7. Transmission Electron Microscopy (TEM) Observation and Selected-Area Electron Diffraction (SAED) Measurement of Precipitated Nodules in a Wide Area

A macroscopic TEM image of a microtome-cut section (Figure 7a) showed an irregularly shaped powder-like precipitate. A higher magnification image (Figure 7b) of the precipitate revealed that it consisted of nanorods (light blue arrows) and bulky materials (black arrows). These nanorods were not observed in the SEM image, probably because of their limited width (less than 10 nm).

The Selected-Area Electron Diffraction (SAED) pattern of the precipitates measured at 800 nm φ showed two Debye rings (Figure 7c), indicating that the crystals in the precipitates were randomly oriented. The rings were 2.89 nm^−1^ (low-intensity ring) and 3.57 nm^−1^ (high-intensity ring) from the center. These characteristic distances were converted to interplanar distances, *d*, of 0.346 and 0.280 nm, respectively. For HAP, the diffraction plane corresponding to *d* = 0.346 nm (with a margin of error of <1.5%) was {002}. For OCP, there were several corresponding planes, such as {121} and {5$\bar{2}$0}. β-TCP also had several corresponding planes, including {1010} and {01$\bar{10}$}. DCPD, however, had no diffraction plane corresponding to a *d* of 0.346 nm.

The *d* of 0.280 nm corresponded to the {211}, {121}, and {112} diffraction planes for HAP, {511}, {7$\bar{1}$0}, etc. for OCP, and {002} for the DCPD. β-TCP had no diffraction plane corresponding to a *d* of 0.280 nm.

The relative intensity of the diffraction plane in each calcium phosphate crystal provided important information for identifying the calcium phosphate phase when the orientations of the crystals were random. The relative intensities of the {002}, {211}, {121}, and {112} diffraction planes of HAP were 49, 100, 43, and 70%, respectively. The diffraction intensities of each plane of the OCP corresponding to *d* = 0.346 or 0.280 nm were all less than 35%. Similarly, low diffraction intensities were also observed for the plane corresponding to *d* = 0.346 nm for β-TCP and the plane corresponding to *d* = 0.280 nm for DCPD. This result strongly suggests that the precipitates were HAP.

Since high-resolution TEM (HR-TEM) observation of the precipitates was difficult due to the overlapping crystals, a lattice image of each crystal in the precipitates was observed using a crushed sample.

### 2.8. High-Resolution TEM (HR-TEM) Observation and Fast Fourier Transform (FFT) Image Analysis of Precipitated Nodules in a Narrow Area

Figure 8a shows a magnified image of nanorods, a major component of the precipitates. Randomly oriented nanorods (white arrows) with a width less than 10 nm were identified, and lattice images were observed for each rod.

HR-TEM observation of a nanorod (Figure 8b) revealed a lattice pattern corresponding to the atomic arrangement. An FFT image of a nanorod (Figure 8c) revealed three periodic directions corresponding to *d* = 0.814, 0.277, and 0.276 nm (white, blue, and yellow arrows, respectively). The *d* of 0.814 nm corresponded to the {26} plane for HAP and the {012}, {10$\bar{2}$}, and {1$\bar{1}$2} planes for β-TCP with a margin of error of <1.5%. In contrast, OCP and DCPD had no planes corresponding to *d* = 0.814 nm. The *d* values of 0.277 and 0.276 nm corresponded with many diffraction planes for HAP, β-TCP, OCP, and DCPD. For HAP, the corresponding planes were {112}, {2$\bar{1}$2}, and {$\bar{2}$12}; for β-TCP, {128}, {3$\bar{2}$8}, etc.; for OCP, {601}, {$\bar{2}$22}, etc., and for DCPD, the *d* of 0.277 nm corresponded to {002}.

The angle of intersection between the directions of the white and blue arrow was 59.0°, and between the directions of the white and yellow arrows was 120.3°. Although the *d*-value analysis indicated the possibility that the nanorod was HAP or β-TCP, comparison of the measured intersection angles with theoretical ones indicated that the nanorod was HAP (Appendix A) The measured *d* values and intersection angles were consistent with the theoretical values with a margin of error of <1.5%. The directions of the white, blue, and yellow arrows corresponded to [100], [112], and $[\bar{2}$12] for HAP. The image in Figure 8b therefore shows the HAP lattice viewed from the [02$\bar{1}$] zone axis. 

We performed similar observations and analyses for ten different nanorods in several areas and found that all of them were HAP.

Figure 9a shows a high-magnification TEM image corresponding to the bulky materials observed in Figure 7b. They consisted of nanoparticles (white arrows) with a diameter less than approximately 10 nm. This observation means that the cell-induced precipitates consisted of a mixture of nanorods and nanoparticles. The 50 nm particles observed by SEM (Figure 6) therefore were secondary particles consisting of smaller nanoparticles.

A HR-TEM image of a nanoparticle (Figure 9b) revealed a lattice arrangement, and a FFT image of nanoparticles (Figure 9c) revealed at least three periodic directions. The *d* values corresponding to each direction (indicated by white, blue, and yellow arrows) were 0.809, 0.385, and 0.285 nm, respectively. The *d* of 0.809 nm corresponded to [26] for HAP, {012}, {10$\bar{2}\},$ and {1$\bar{1}$2} for β-TCP with a margin of error of <1.5%. OCP and DCPD had no planes corresponding to *d* = 0.809. The *d* of 0.385 nm corresponded to {1$\bar{2}$1}, {111}, and {$\bar{2}$11} for HAP with a margin of error of <1.5%. OCP corresponded to several planes, such as {1$\bar{2}\bar{1}$} and {2$\bar{2}\bar{1}$}. β-TCP and DCPD had no corresponding planes. The *d* of 0.281 nm corresponded to many planes for both HAP and OCP: {3$\bar{2}$1}, {21$\bar{1}$}, etc. for HAP, and {130}, {420}, etc. for OCP. DCPD corresponded to {002}, whereas β-TCP had no corresponding plane. The *d*-value analysis indicated that only HAP had planes corresponding to three directions. The intersection angle between the directions of the white (*d* = 0.809 nm) and blue (*d* = 0.385 nm) arrows was 90.7°, and between the directions of the white and yellow (*d* = 0.281 nm) arrows was 46.9°. These angles corresponded to those between [100] and [1$\bar{2}$1] and between [100] and [3$\bar{2}$1] for HAP, respectively. The measured *d* values and intersection angles between each plane were consistent with the theoretical values (with a margin of error <1.5%; see Appendix A). The image in Figure 9b therefore corresponded to the HAP lattice viewed from the [012] zone axis. We performed similar observations and analyses for ten different nanoparticles in several areas and found that all of them were HAP.

### 2.9. Element Content in Precipitated Nodules Measured Using Scanning TEM Energy-Dispersive X-ray Spectroscopy (STEM-EDS)

Figure 10 shows a low-magnification (approximately 12 μm square) TEM image (a) and two-dimensional elemental mappings of Ca (b), P (c), and N (d) in the area in (a). Ca and P were observed in only the precipitates, and N was mostly observed in the cell region. A small amount of N was detected in the precipitate region; it was probably due to cell contamination. The energy-dispersive X-ray spectroscopy (EDS) spectrum (Figure 10e, magenta curve) corresponded to the region in Figure 10a where the average Ca/P atomic % ratio was 1.41 ± 0.01. This Ca/P ratio increased slightly with a decrease in the area size and reached equilibrium when the area size was less than a few μm. The blue and green curves show the spectra measured for 1.5 μm and 200 nm squares, the corresponding Ca/P ratios of which were 1.44 ± 0.01 and 1.45 ± 0.01. The detected elements (except C, O, N, and Cu) and their contents for the 200 nm square are shown in Appendix A. 

We performed EDS measurements on five different areas in the 1.5 μm square and determined that the average Ca/P atomic % ratio was 1.45 with a deviation of ± 0.02.

## 3. Discussions

Cytokines such as BMP and TGF-β play important roles in the regulation of odontoblast differentiation [14,15,27,28]. When we established the PPU-7 cell line, we demonstrated that both BMP2 and TGF-β1 significantly affected cell proliferation, cell population doubling time, mRNA expression of dentin sialophosphoprotein (DSPP), which is one of the differentiation marker for odontoblasts, and ALP activity in the PPU-7 cell line. These findings are in agreement with the reports described above. In addition, considering the inherent ALP activity in the PPU-7 cell line, our data suggest that some level of ALP activity is required for the induction of the differentiation of PPU-7 cells into hard tissue-forming cells by BMP2 and/or TGF-β1. 

In vitro study using human mesenchymal stem cells (hMSCs) shows that MDZ inhibits ALP activity and calcium deposition in hMSCs, suggesting a suppressive effect on osteogenic differentiation [23]. However, little was known about the effect of MDZ on the properties of dental pulp tissues. We therefore hypothesized that MDZ affects odontoblastic differentiation rather than the osteogenic differentiation of dental pulp stem cells. We demonstrated that the addition of MDZ-only to PPU-7 cells dramatically enhanced ALP activity and mineralization induction. This finding suggests that MDZ possesses the ability to induce the differentiation of PPU-7 cells into hard tissue-forming cells. Moreover, the results led us to investigate the combination of MDZ and cytokines, namely, BMP2 and TGF-β1.

In general, many drugs express their drug efficacy through a receptor. MDZ controls the GABA, which is the main inhibitory neurotransmitter. The GABAA receptor is a multi-subunit chloride channel that mediates the fastest inhibitory synaptic transmission in the central nervous system. The benzodiazepine class, which includes MDZ, enhances the effect of the neurotransmitter GABA at the GABAA receptor [17]. The GABAA receptor has at least 19 different subunits and shows different susceptibility to drugs depending on the combination of units [29,30]. The GABAA receptor beta 1 subunit (GABRB1) is strongly expressed in the human coronal pulp and permanent predentin/odontoblasts [31,32]. Although we demonstrated that MDZ-only treatment enhanced ALP activity and mineralization induction, combinations of MDZ with BMP2 or TGF-β1 tended to decrease both of them. Our findings suggest that the pharmacological interaction of the combination of MDZ with BMP2 or TGF-β1 reduces the susceptibility and/or reactivity of GABRB1.

Dental pulp stem cells possess a pluripotency for various types of specialized cells, such as neuron, cardiomyocytes, chondrocytes, odontoblasts, osteoblasts, and liver cells [33,34]. The starting population of these cells and their differentiated progeny are evaluated by analyzing the expression of specific gene markers. DSPP is an important marker of odontoblastic differentiation. We previously generated two amplification products from the full-length *DSPP* (*DSPP-v1*) and the DSP-only (*DSPP-v2*) transcripts and found that both the *DSPP-v1* and *DSPP-v2* products were predominantly observed in odontoblasts, whereas only trace expression of the *DSPP-v1* transcript was detected in dental pulp [35]. Our previous study showed that in addition to these two DSPP splice variants, matrix metalloproteases such as *MMP2* and *MMP20* can become odontoblast differentiation markers [36]. We demonstrated that the mRNA levels of *DSPP-v1*, *DSPP-v2* and *MMP2* were dramatically enhanced by MDZ at day seven. In contrast, mRNA expression levels of the osteoblastic differentiation markers OC and *RUNX2*, and the chondrogenic differentiation markers Col II and *ACAN* decreased over time regardless of the presence or absence of MDZ. These findings suggest that MDZ promotes the differentiation of PPU-7 cells into odontoblasts.

Alizarin Red S staining has been widely used to evaluate calcium deposits in cell culture [37], whereas von Kossa staining has generally been used to detect both phosphates and carbonates in calcium deposits [38]. We demonstrated that the formation of precipitated nodules in a mineralization-inducing culture medium in the presence or absence of MDZ was evident on the plate by both staining methods and that the MDZ-only treatment showed the highest ability to induce calcification. These findings suggest that the precipitated nodules possess calcium carbonate-based or calcium phosphate-based components and that the use of MDZ-only is the most effective in inducing mineralization in the PPU-7 cell line. 

DPP is a highly phosphorylated protein in dentin with an isoelectric point near 1.1. Due to the biased amino acid composition and extensive acidity (35–40% phosphoserine and 40–50% aspartic acid) [39], DPP stains very strongly with Stains-all but does not stain with Coomassie Brilliant Blue. Genetic studies have shown that for porcine DPP, the length variations in the DPP code are closely correlated with the length variations observed at the protein level [40]. Therefore, DPP samples separately isolated from 22 pigs appear as single or doublet bands migrating between 96 kDa and 100 kDa on SDS-PAGE [40]. Our previous study showed that full-length *DSPP* containing DPP is an odontoblast-predominant transcript and that little DPP is present in porcine dental pulp, suggesting that DPP is a dentin-specific protein [35]. In this study, we demonstrated that DPP doublet bands were present in the precipitated nodules in PPU-7 cells cultured with a mineralization-inducing culture medium and that the addition of MDZ increased the intensity of the DPP doublet bands. Our data suggest that the precipitated nodules possessed dentin-like characteristics. 

The HR-TEM observations and FFT analyses indicated that the nanorods and nanoparticles were HAP. To support this conclusion, SAED simulation of a HAP crystal viewed from a particular zone axis was performed using the ReciPro application. Appendix A shows the SAED patterns of a HAP crystal viewed from {02$\bar{1}$} (Appendix A) and {012} (Appendix A). The pattern corresponding to the {02$\bar{1}$} direction (dotted hexagon) was essentially the same as the FFT image in Figure 8c. The diffraction spots corresponding to {012} and {$\bar{1}$12} were not clearly visible in the FFT image, probably due to the width of the nanorods in those directions not having sufficient periodicity in the image. The simulated SAED corresponding to the {012} direction was consistent with the FFT image in Figure 9c, indicating that the nanorods and nanoparticles were indeed HAP. The two Debye rings observed in the wide-area SAED pattern (Figure 7c) revealed the most likely *d* values in randomly oriented HAP, and the XRD pattern of the precipitates showed peaks corresponding to those of HAP. These findings indicated that the precipitates consisted of a single phase of low-crystalline HAP. The STEM-EDS results showed that the HAP in the precipitates was calcium deficient.

In our previous study, an enamel-like HAP nanorod array was constructed on an amorphous calcium phosphate (ACP) substrate consisting of nanoparticles (<80 nm diameter) that had been compression molded [41]. When the substrate was immersed in a calcium phosphate solution without cells under pseudo-physiological conditions, ACP nanoparticles close to the substrate surface quickly transformed into HAP ones and grew perpendicularly as rods due to geometric selection. During this reaction, ACP nanoparticles located in the middle to deep regions of the substrate were transformed into low-crystalline HAP. We found that the macroscopic and microscopic morphologies of cell-induced HAP closely resembled those of HAP transformed from ACP nanoparticles. Appendix A shows the TEM images, SAED patterns, and STEM-EDS spectra for both types of HAP crystal. The macroscopic morphology of the grown material was fibrous for both types (Appendix A) and consisted of a mixture of nanorods and nanoparticles (Appendix A). Regarding the morphology in elemental crystals and their aggregates, the two types of HAP crystal were similar. The SAED patterns for the grown materials measured at 800 nm φ were also similar (Appendix A). The average Ca/P atomic % ratio of cell-induced HAP (Ca/P = 1.45) was slightly less than that of HAP transformed from ACP nanoparticles (Ca/P = 1.50) (Appendix A) and that of human dentin [42]. One reason for this difference is the substitution of several cations, such as Zn and Mg, at Ca-ion sites in the cell-induced HAP, which occurred due to the complex composition of the initial growing fluid. Based on the content of each element shown in Appendix A, the Ca/(Zn + Na + Mg + Si) atomic % ratio was approximately 7.48, meaning that the x component in Ca_10−x_(Zn, Na, Mg, Si)_x_(PO_4_)_6_(OH)_2_ was approximately 1.18. Therefore, the calculated Ca/P molar ratio of the cell-induced HAP was 1.47, which was close to the measured ratio for the precipitates. 

The morphological, structural, and chemical similarities between the cell-induced HAP and the ACP-transformed HAP strongly suggest that the cells, which had high ALP activity, produced locally concentrated phosphate ions, which nucleated ACP nanoparticles in the initial reaction stage. These particles subsequently transformed into low-crystalline HAP. Biological analysis after HAP precipitation showed that the initial pulpal cells differentiated into odontoblast cells and that the present system simulated the dentin matrix formation process, suggesting that the HAP in tooth dentin is formed by phase transformation from ACP nanoparticles. Indeed, a TEM image of human molar dentin (Appendix A) showed that it consisted of nanorods and nanoparticles, the same as for the images in Appendix A.

## 4. Materials and Methods 

These studies have received approval from the Institutional Ethics Review Committee of the Tsurumi University School of Dental Medicine (Project identification code #1318, 1 December 2015). Additionally, all animal experiments were approved by the Institutional Animal Care Committee and the Recombination DNA Experiment and Biosafety Committee of the Tsurumi University School of Dental Medicine. We have confirmed that all experiments were performed in accordance with relevant guidelines and regulations.

### 4.1. Alkaline Phosphatase (ALP) Activity Assay of the PPU-7 Cell Line

The PPU-7 cell line was established from porcine dental pulp cells by our group (see Appendix A). We measured ALP activity in each well as described previously [43]. The cells were plated on a 96-well plate at a density of 3.16 × 10^4^ cells/cm^2^ and were cultured in the standard medium for 24 h. The medium was changed to growth medium supplemented with 0, 2.5, 5, or 10 μM of MDZ with 500 ng/mL of rhBMP2 (R&D Systems, Minneapolis, MN, USA) or with 1 ng/mL of rhTGF-β1 (Cell Signaling Technology, Danvers, MA, USA). After 72 additional hours of incubation, the cells were washed once with PBS, and ALP activity was assayed using 10 mM p-nitrophenylphosphate as the substrate in 100 mM 2-amino-2-methyl-1,3-propanediol-HCl buffer (pH 10.0) containing 5 mM MgCl_2_ and incubated for 10 min at 37 °C. Adding 0.2 M NaOH quenched the reaction, and the absorbance at 405 nm was read on a plate reader.

### 4.2. Alkaline Phosphatase Staining

The cells were spread on a 24-well plate at density of 3.16 × 10^4^ cells/cm^2^. After incubation for 2 days, the cells were rinsed twice with PBS, fixed with 10% formaldehyde for 30 min, stained with 0.1 mg/mL of naphthol AS-MX phosphate (Sigma-Aldrich, St. Louis, MO, USA), 0.5% N,N-dimethylformamide, Fast Blue BB salt (Sigma-Aldrich), and 2 mM MgCl_2_ in 0.1 M Tris–HCl buffer (pH 8.5) for 30 min at room temperature, and then washed with dH_2_O and photographed.

### 4.3. Gene Expression of the PPU-7 Cell Line

The cells were extracted with an RNA extraction reagent (Isogen, Nippon Gene Co., Ltd., Tokyo, Japan). The purified total RNA (2 μg) was reverse transcribed; the reaction mixture consisted of SYBR Green PCR master mix (Roche), supplemented with 0.5 µM forward and reverse primers and 2 µL of cDNA as template. The specific primer sets were designed using Primer-BLAST software (URL: http://www.ncbi.nlm.nih.gov/tools/primer-blast). The specific primer sets and running conditions are shown in Appendix A. *GAPDH* was used as the reference gene. Each ratio was normalized the relative quantification data of *DSPP-v1*, *DSPP-v2*, *MMP2*, OC, *RUNX2*, Col II and *ACAN* in comparison to a reference gene (*GAPDH*), generated on the basis of a mathematical model for relative quantification in qPCR system. All values were represented as means ± standard error. Statistical significance (*) was determined using a Steel’s test, whereas (**) was determined using a Mann–Whitney U-test. In all cases, *p* < 0.05 was regarded as statistically significant.

### 4.4. Detection of Precipitated Nodules in PPU-7 Cells

PPU-7 cells were grown on a 12-well plate at an initial density of 3.16 × 10^4^ cells/cm^2^. After incubation for 24 h, the medium was changed to a mineralization-inducing medium containing 10 mM β-glycerophosphate and 50 μM ascorbic acid. The cells were cultured for up to 7 days. Mineralization was visualized by Alizarin Red S and von Kossa staining. After fixation with 4% paraformaldehyde neutral buffer solution for 30 min, the cells were stained with 1% Alizarin Red S (Sigma-Aldrich) solution for 10 min, then washed with distilled water and photographed. Alternatively, silver staining was performed using a von Kossa staining kit (Polysciences, Inc., Warrington, PA, USA) according to the manufacturer’s instructions.

PPU-7 cells were grown on a 24-well plate at an initial density of 3.16 × 10^4^ cells/cm^2^. After incubation for 24 h, the medium was changed to a mineralization-inducing medium. The cells were cultured for up to 5 days. Each well on the plates was rinsed with PBS, and the calcium was dissolved in 0.5 mL of 0.5 N HCl by gentle rocking for 1 h. The calcium concentration in the eluate was spectrophotometrically determined at 570 nm by following the color development with a calcium assay kit (Calcium C-test Wako, Wako Pure Chemical Industries, Ltd.). All values were normalized against the cultivation area.

### 4.5. Statistical Analysis

For ALP assays and qPCR and calcium analyses, all values are presented as the mean ± standard error. Statistical significance (*) was determined using the Mann–Whitney U-test for qPCR analysis and a nonparametric Steel’s test for the ALP assay and calcium analysis. *p* < 0.05 was regarded as statistically significant for Steel’s test, and *p* < 0.01 or *p* < 0.05 was regarded as statistically significant for the Mann–Whitney U-test.

### 4.6. Characterization of Proteins in the Precipitated Nodules from PPU-7 Cells

The PPU-7 cell line was grown in standard medium or in mineralization-inducing culture medium on the 6 cm culture dish at an initial density of 3.16 × 10^4^ cells/cm^2^ for 14 days, and the cell sheet was scraped off with a cell scraper. The cell sample was suspended with 50 mM Tris–HCl/4M guanidine buffer (pH 7.4) containing Protease Inhibitor Cocktail Set III [1 mM 4-(2-aminoethyl) benzenesulfonyl fluoride hydrochloride (AEBSF), 0.8 mM aprotinin, 50 mM bestatin, 15 mM E-64, 20 mM leupeptin, and 10 mM pepstatin (Calbiochem EMD Chemicals Inc., Gibbstown, NJ, USA)] and 1 mM 1,10-phenanthroline (Sigma-Aldrich), and was homogenized using a Polytron (Capitol Scientific, Inc., Austin, TX, USA) homogenizer for 30 s at half speed. Insoluble material was pelleted by centrifugation (15,900g) and the supernatant (G1 extract) was stored at −80 °C. The guanidine-insoluble material was extracted against 0.17 N HCl and 0.95% formic acid containing 10 mM benzamidine (Sigma-Aldrich), 1 mM PMSF, and 1 mM 1,10-phenanthroline using the homogenizer. Following centrifugation, the acid-soluble supernatant (HF extract) was stored at −80 °C. The pellet was extracted with 50 mM Tris–HCl/4M guanidine buffer (pH 7.4) containing Protease Inhibitor Cocktail Set III and centrifuged, and the supernatant (G2 extract) was stored at −80 °C. The G1, HF, and G2 extracts were desalted with an Amicon Ultra centrifugal filter (0.5 mL, MW = 3000 cut off) (Merck Millipore, Darmstadt, Germany). Following the quantitative analysis of total protein in each extract using a Pierce 660 nm Protein Assay Reagent (Thermo Fisher Scientific, Waltham, MA, USA), the amount of total protein was normalized by dividing the amount of total protein in control (i.e., PPU-7 cell line was cultured in the standard medium), and characterized by sodium dodecyl sulfate–polyacrylamide gel electrophoresis (SDS-PAGE).

SDS-PAGE was performed using a 15% e-PAGEL mini gel (ATTO Corporation, Tokyo, Japan). The gel was stained with SimplyBlue SafeStain (Invitrogen) or Stains-all stain (Sigma-Aldrich). The apparent molecular weights of the protein bands were estimated by comparison with the SeeBlue Plus2 Pre-Stained standard (Life Technologies/Invitrogen). 

### 4.7. Degree of Supersaturation with Respect to Each Calcium Phosphate Phase

The calcium phosphate phases that form under physiological conditions are DCPD, OCP, β-TCP, and HAP. We calculated the degree of supersaturation, σ, of the initial culture fluid with respect to each phase using:σ = (IP/K_sp_)^1/n^ − 1 (n = 2, 16, 5, and 18 for DCPD, OCP, β-TCP, and HAP, respectively)(1)
where IP is the ionic activity of the solution and K_sp_ is the solubility product constant of the material at 37 °C [44]. The concentration of each ion in the fluid and the degree of supersaturation are given in Appendix A.

### 4.8. XRD

Microbeam XRD (RAPID, Rigaku) with monochromated Cu-Kα radiation was used to characterize the precipitates at 50 kV and 30 mA. The incident beam was focused to a diameter of approximately 100 μm and irradiated perpendicular to the surface of the cell sheet. The diffraction was recorded on an imaging plate, and the digital data on the plate were converted to the intensity versus 2θ relationship using DISPLAY software (Rigaku). The diffraction peak positions were referenced to those of different calcium phosphate phases in the JCPDS cards (DCPD: Card 11–293, OCP: Card 26–1056, β-TCP: Card 9–169, HAP: Card 9–432).

### 4.9. SEM

Field-emission scanning electron microscopy (FE-SEM, JSM-7000F, JEOL, Ltd.) with an acceleration voltage of 5 kV was used to observe cross-sectional samples. The samples were coated with an osmium film (approximately 3 nm thick) before observation.

### 4.10. TEM Observation and Elemental Analysis Using STEM-EDS

Two types of samples were prepared for TEM observations. Ultramicrotome-cut samples were prepared by embedding cell sheets with precipitates in epoxy resin and placing them in a dark room for 48 h at room temperature. After the resin completely filled the gaps (precipitate regions) between cell sheets, the resin was solidified using ultraviolet irradiation. The solidified block was then sliced using a diamond knife into samples with a thickness of 70–100 nm. The samples were placed on a TEM grid with a Cu mesh for observation. Samples crushed in an agate mortar were dispersed in 99.5% ethanol solution, which was ultrasonically stirred for 5 min at 40 kHz. An aliquot of the stirred solution was placed on a TEM grid with a Cu mesh and allowed to dry naturally in air. An analytical TEM (Tecnai Osiris, FEI Co.) with an acceleration voltage of 200 kV was used to observe the samples. SAED measurement with a typical area of 800 nm or 200 nm φ was simultaneously performed to characterize the material. 

Two-dimensional elemental mapping and quantitative analysis of the content of each element were performed using a Super-X EDS system in the TEM. The probe diameter used for analysis was approximately 0.5 nm, and the amplitude was approximately 0.55 nA. The beam residence time at each measurement position was 10 μs, and the analysis was completed within 5 min. The relative content of each element was calculated from the energy-dispersive spectra by using peak deconvolution. 

## 5. Conclusions

Drug repositioning is the process of discovering, validating, and marketing previously approved drugs for new indications. Due to the promise of reduced costs and expedited approval schedules, the research field of drug repositioning has been attracting attention [45]. To find candidates for drug repositioning, a standard database consisting of both approved and failed drugs has been developed as a web application [46]. Without using the database, the present study demonstrated that MDZ enhances the differentiation of PPU-7 cells to odontoblast and promotes the formation of dentin-like hydroxyapatite. Further studies are required to elucidate the pharmacokinetic and pharmacological efficacy of MDZ in animal experiments. In the dental field, these findings support the repositioning of MDZ to promote dentin regeneration for endodontic treatments, such as pulp capping. Moreover, these findings support advancing research from pig to human experimental models using human DPSCs to discover MDZ’s potential, not only for future dental treatments, but also for organ regenerative medicine.

## Figures and Tables

**Figure 1 ijms-20-00670-f001:**
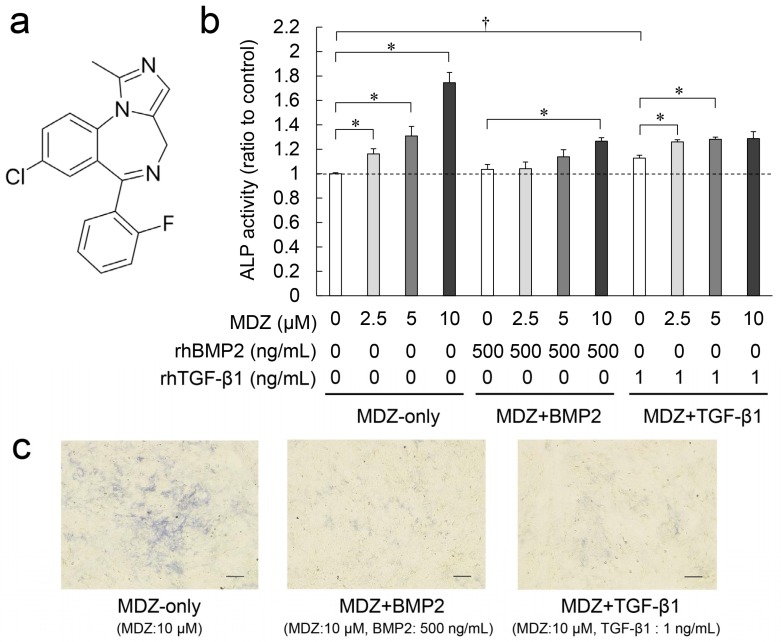
Combined effect of midazolam (MDZ) with bone morphogenetic protein 2 (BMP2) or transforming growth factor beta 1 (TGF-β1) on alkaline phosphatase (ALP) activity in the porcine dental pulp-derived (PPU-7) cell line. (**a**) Structure of MDZ. (**b**) ALP-inducing activity of midazolam without BMP2 or TGF-β1 (MDZ-only), midazolam with BMP2 (MDZ and BMP2) and midazolam with TGF-β1 (MDZ and TGF-β1) in PPU-7 cells. ALP activities are indicated as increasing or decreasing ratios relative to the level of the control (i.e., 0 μM MDZ, 0 ng/mL BMP2, 0 ng/mL TGF-β1), which was set at 1 (dotted line). Values are the means ± standard error of six culture wells. Significant differences are indicated by an asterisk (* *p* < 0.05, Steel’s test) or a dagger (^†^
*p* < 0.05, Mann–Whitney U-test). (**c**) ALP staining for PPU-7 cells cultured with MDZ-only, MDZ and BMP2, MDZ and TGF-β1 (Scale bar: 200 μm).

**Figure 2 ijms-20-00670-f002:**
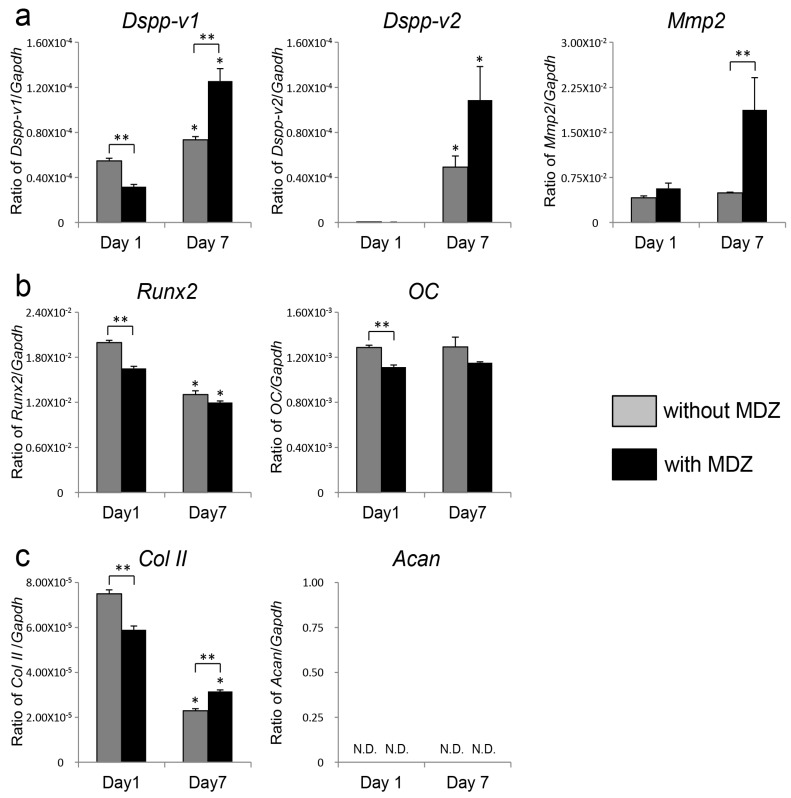
Effect of MDZ on temporal changes in gene expression of the PPU-7 cell line. The mRNA expression by quantitative polymerase chain reaction (qPCR) analysis of (**a**) odontoblastic differentiation markers, i.e., *DSPP*-variant 1 (*DSPP-v1*), *DSPP*-variant 2 (*DSPP-v2*) and matrix metalloprotease 2 (*MMP2*); (**b**) osteoblastic differentiation markers, i.e., osteocalcin (OC) and runt-related transcription factor 2 (*RUNX2*); and (**c**) chondrogenic differentiation markers, i.e., type II collagen (Col II) and aggrecan (*ACAN*). Each ratio was normalized to glyceraldehyde-3-phosphate dehydrogenase (*GAPDH*) as a reference gene, and the relative quantification data of *DSPP-v1*, *DSPP-v2*, *MMP2*, OC, *RUNX2*, Col II and *ACAN* in PPU-7 cell line were generated on the basis of a mathematical model for relative quantification in a qPCR system. Values are the means ± standard error of six culture wells. The asterisk (*) on the bar graph indicates a significant difference (*p* < 0.05, Steel’s test) between day one and day seven. The double asterisk (**) on the bar graph indicates a significant difference (*p* < 0.05, Mann–Whitney U-test) between cells cultured with and without MDZ.

**Figure 3 ijms-20-00670-f003:**
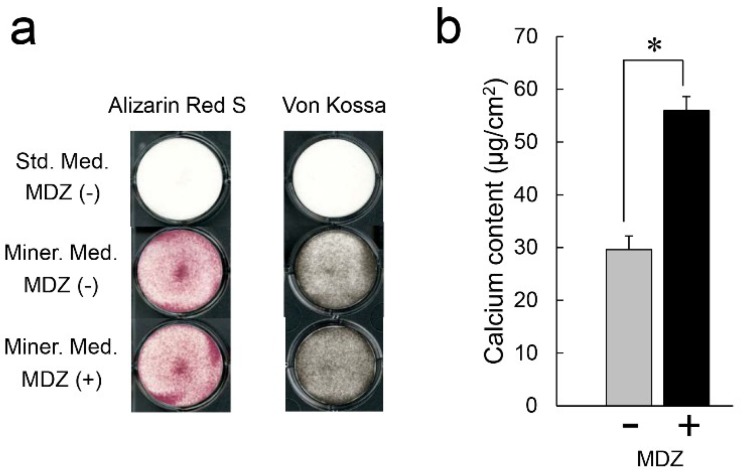
Effect of MDZ on nodule formation in the PPU-7 cell line. Nodule cultures were stained with (**a**) Alizarin Red S (left) and von Kossa (right) staining on day seven. In contrast to PPU-7 cells not subjected to mineralization induction, PPU-7 cells cultured in mineralization-inducing culture media clearly exhibited nodule formation regardless of the addition of MDZ. (**b**) Calcium contents in PPU-7 cells were determined on day five after the mineralization induction. Values are the means ± standard error of six culture wells. The asterisk (*) on the bar graph indicates a significant difference (*p* < 0.05, Mann–Whitney U-test) between the cells incubated with and without MDZ. Std. Med.: Standard culture medium, Miner. Med.: Mineralization-inducing culture medium.

**Figure 4 ijms-20-00670-f004:**
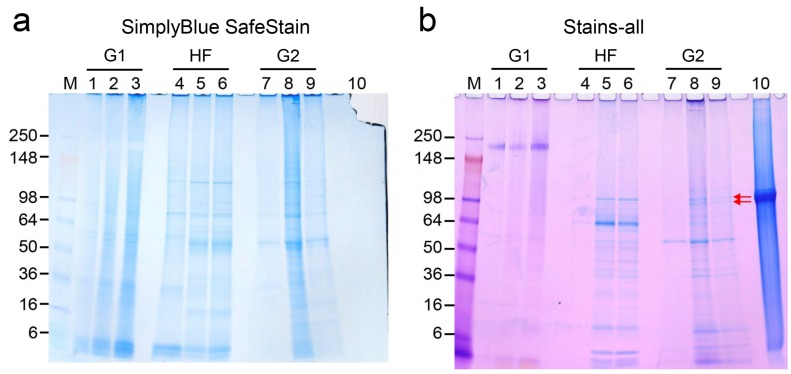
Sequential extraction of proteins from precipitated nodules induced by PPU-7 cells. Precipitated nodules formed in PPU-7 cells cultured in mineralization-inducing culture medium with (MDZ(+)) or without (MDZ(−)) MDZ were sequentially extracted by tris-guanidine (G1 extraction), formic acid–HCl (HF extraction) and tris-guanidine again (G2 extraction). The PPU-7 cells cultured with only standard medium were also extracted as a control. Analysis of the G1, HF, and G2 extracts by 5–20% gradient SDS-PAGE were stained with (**a**) SimplyBlue SafeStain and (**b**) Stains-all stain. Dentin phosphoprotein (DPP) doublet bands in lanes 5, 6, 8, 9, and 10 detected by Stains-all staining are indicated with a red arrow. Lanes 1, 4 and 7: Control; lanes 2, 5 and 8: MDZ(+); lanes 3, 6 and 9: MDZ(-); lane 10: DPP purified from porcine dentin. M: Molecular weight marker (SeeBlue Plus2 Pre-Stained standard).

**Figure 5 ijms-20-00670-f005:**
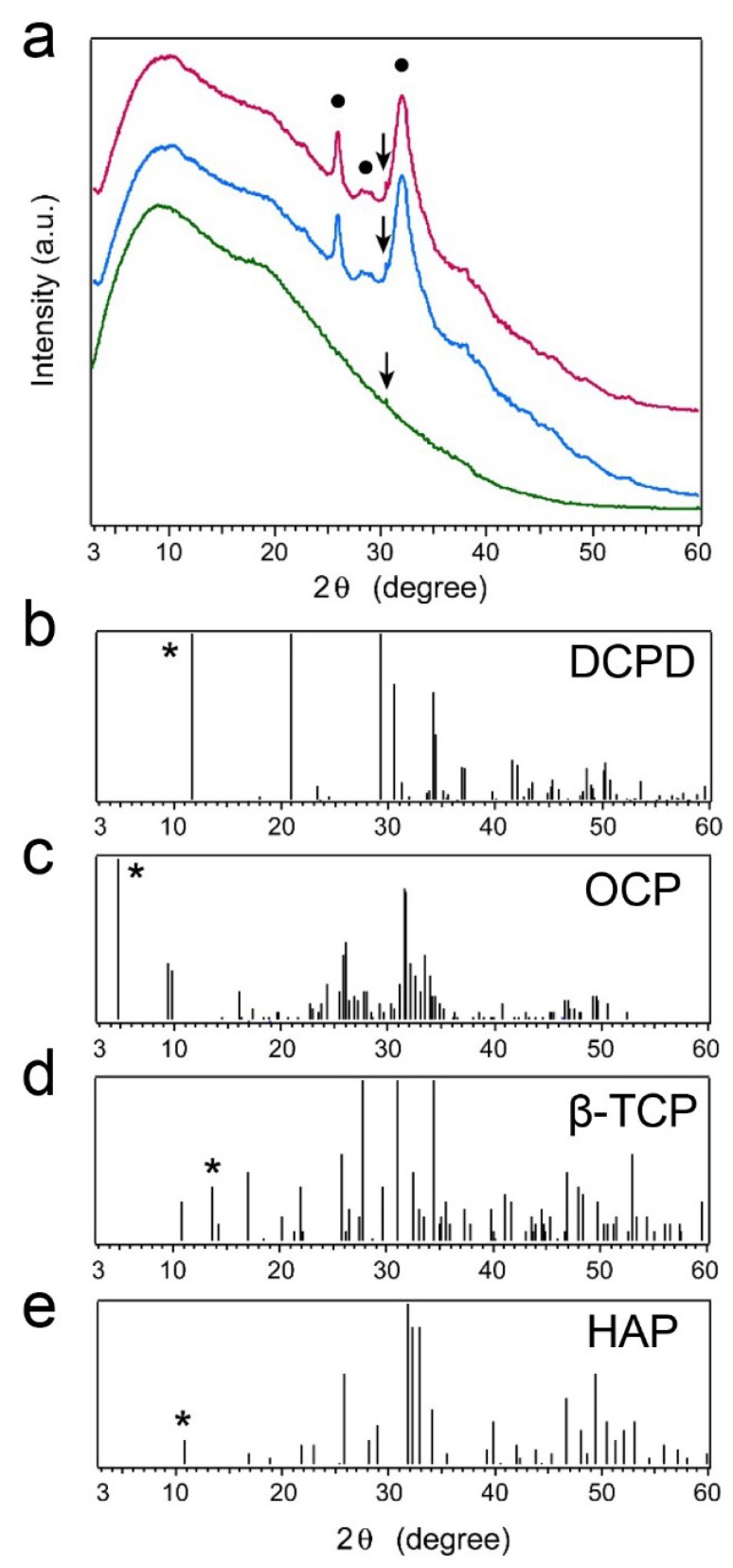
(**a**) Microbeam XRD patterns for samples grown with (magenta curve) and without (blue curve) MDZ in culture solution and for a cell sheet (green curve) for reference. Peaks attributed to the precipitates are indicated by black circles; spike peaks attributed to damage to the imaging plate are indicated by arrows. Ideal XRD patterns for (**b**) dicalcium phosphate dihydrate (DCPD), (**c**) octacalcium phosphate (OCP), (**d**) β-tricalcium phosphate (β-TCP), and (**e**) hydroxyapatite (HAP) derived using 2θ versus diffraction–intensity relationships in the corresponding JCPDS cards. The peaks used to identify each calcium phosphate phase are indicated by asterisks.

**Figure 6 ijms-20-00670-f006:**
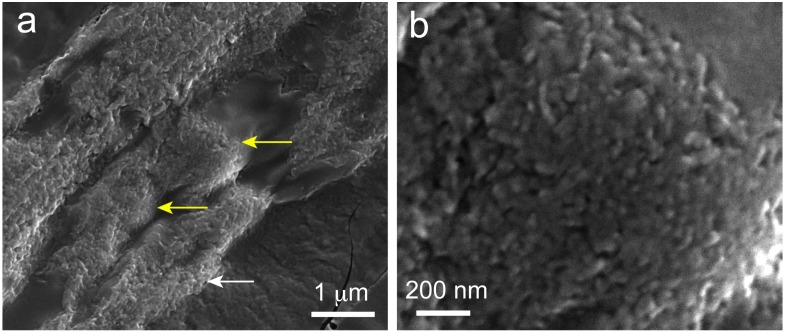
SEM images of a cross-sectional sample. (**a**) Low-magnification image showing irregularly shaped precipitates (white arrow) and ball-like ones (yellow arrows). (**b**) High-magnification images of ball-like precipitate showing nanoparticles sized approximately 50 nm.

**Figure 7 ijms-20-00670-f007:**
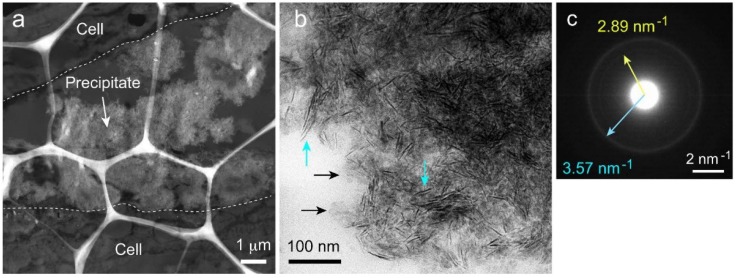
Wide-area TEM images of precipitates. (**a**) Low-magnification image of a microtome-cut sample. The dotted line shows the boundary between the cell sheet and precipitate region. (**b**) High-magnification image of precipitates. Many nanorods (light blue arrows) and bulky materials (black arrows) were observed. (**c**) Selected-Area Electron Diffraction (SAED) pattern measured at 800 nm φ. Two Debye rings (low and high intensities) were observed.

**Figure 8 ijms-20-00670-f008:**
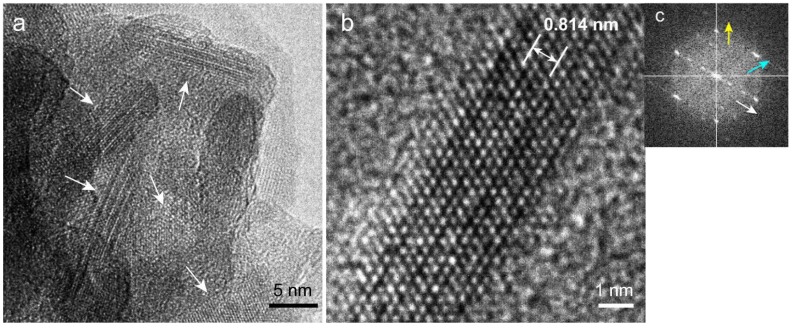
Narrow-area TEM images of nanorods. (**a**) Aggregation of nanorods. The orientation of each rod was random, and lattice fringes were observed in the rods. (**b**) HR-TEM image of a rod. The lattice image of the rod is visible. (**c**) FFT image of a rod. Three directions of the crystal planes are indicated by white, blue, and yellow arrows. The interplanar distance corresponding to the direction of the white arrow (0.814 nm) is superimposed in (**b**).

**Figure 9 ijms-20-00670-f009:**
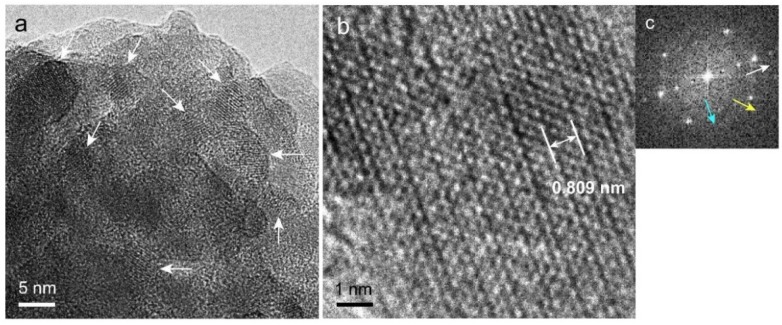
Narrow-area TEM images of bulky precipitates. (**a**) The precipitates consisted of nanoparticles sized less than approximately10 nm. (**b**) HR-TEM image of a nanoparticle. (**c**) FFT image of a nanoparticle. Three directions of crystal planes are indicated by white, blue, and yellow arrows. The interplanar distance corresponding to the direction of the white arrow (0.809 nm) is superimposed in (**b**).

**Figure 10 ijms-20-00670-f010:**
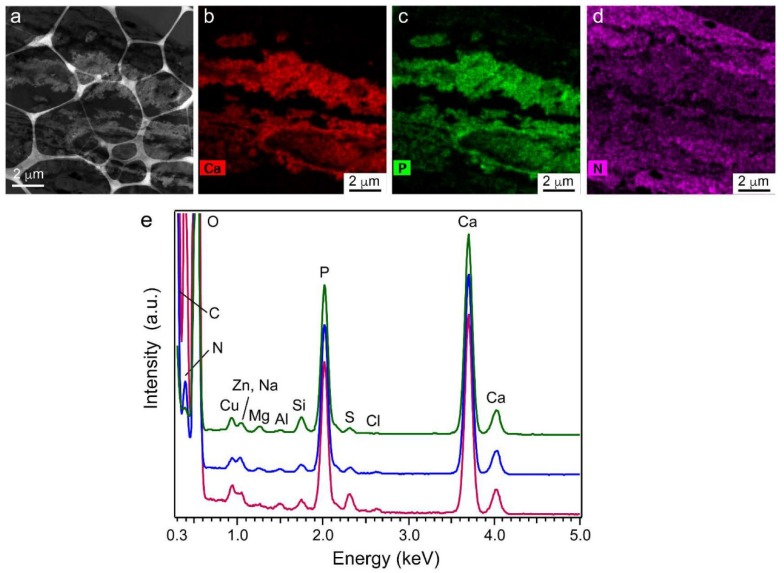
Scanning TEM energy-dispersive X-ray spectroscopy (STEM-EDS) analysis results for precipitates. (**a**) Low-magnification TEM image for energy-dispersive X-ray spectroscopy (EDS) analysis. Two-dimensional elemental mappings of (**b**) Ca, (**c**) P, and (**d**) N. (**e**) STEM-EDS spectra for three measured areas: approximately 12 μm, 1.5 μm, and 200 nm (magenta, blue, and green curves, respectively). Cu is attributed to the TEM grid, and Al is equipment specific. Each spectrum was normalized by the intensity of Ca at 3.7 keV.

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
