# Peer review of "Potential for Drug Repositioning of Midazolam for Dentin Regeneration"

_ijms, 2019, doi:10.3390/ijms20030670_

Reviewer 1 Report

Dear Authors 

This manuscript looks quite and attractive to this special issue for the readers. 

In line 36-52: These paragraphs are quite limited scientific information. These lines lacked the art of scientific write-up. Here are few references will enhance scientific information;

a)  https://link.springer.com/article/10.1007/s13770-015-0030-6 

b) https://www.ncbi.nlm.nih.gov/pmc/articles/PMC5745671/ 

c) https://www.ncbi.nlm.nih.gov/pmc/articles/PMC5725894/ 

In line number 53-62: very limited information on the Midazolam (MDZ). Authors have to provide more few lines on it and link with their experiment. 

In Result, section authors have to rephrase the heading 2.2 scientifically. The interpretation of the results is quite complicated for the readers. 

Figure-6: a, b SEM images are quite visible but c is not in good resolution. Authors have to replace it or omit. 

In conclusion, authors have to share some lines on future direction. 

Author Response

The authors wish to thank the first reviewer for their helpful and constructive comments and helpful suggestions, especially with regard to improving our use of specific terms and sentence structure.

Response for the first reviewer

This manuscript looks quite attractive to this special issue for the readers.

The authors wish to thank the first reviewer for their helpful and constructive comments and helpful suggestions, especially with regard to improving our use of specific terms and sentence structure. Our response to the reviewer’s comments is as follows:

In line 36-52: These paragraphs are quite limited scientific information. These lines lacked the art of scientific write-up. Here are few references will enhance scientific information;

a)  https://link.springer.com/article/10.1007/s13770-015-0030-6

b) https://www.ncbi.nlm.nih.gov/pmc/articles/PMC5745671/

c) https://www.ncbi.nlm.nih.gov/pmc/articles/PMC5725894/

A. We have added the following general information in lines 37 to 41 of the Introduction: “Tissue engineering is a multidisciplinary science. Applications of tissue engineering are founded on three components; a cell source, a scaffold and bioactive molecules. In the field of dental tissue engineering, various dental soft and hard tissues have been regenerated in vitro using stem cells [1]. Dental pulp stem cells (DPSCs) have been isolated with various technique and used for studies related to the cell differentiation potential and scaffold for tissue regeneration [2,3]”. This addition cites the references kindly provided by the reviewer.

In line number 53-62: very limited information on the Midazolam (MDZ). Authors have to provide more few lines on it and link with their experiment.

A. We corrected the introduction in lines 55 to 72 to read: “Drug repositioning is where new therapeutic applications are identified for existing drugs. In addition to the studies on dental pulp cells expressing cytokines, such as BMP, FGF and TGF-β, existing drugs for treating Alzheimer’s disease have been reported to promote dentin regeneration [16]. In Japan, Alzheimer’s drugs such as donepezil hydrochloride (Aricept) have been widely used as acetylcholinesterase inhibitors that increase acetylcholine, an excitatory neurotransmitter in the vertebrate nervous system. This study focuses on repositioning midazolam (MDZ), which controls gamma-aminobutyric acid (GABA), the principal inhibitory neurotransmitter in the mammalian central nervous system. Midazolam (MDZ) is a chemically synthesized imidazobenzodiazepine derivative that possesses pharmacological effects as a hypnotic, sedative, and anesthetic, has anti-anxiety and anti-convulsant properties, and act as muscle relaxant [17]. In the dental field, MDZ has been used mainly as a sedative prior to dental anesthesia. The intravenous MDZ formulation has been recommended as a first-line drug for treating convulsive status epilepticus [18]. Intravenous MDZ has not been approved to treat status epilepticus in most countries, but it has been used off-label for patients in Japan. In tumor and cancer cells, MDZ induces cellular apoptosis by regulating the caspase pathway, endoplasmic reticulum stress, autophagy and the cell cycle [19-22]. Few studies have focused on the effect of MDZ on dental pulp cells, although an in vitro study using human mesenchymal stem cells (hMSCs) has shown that MDZ inhibits ALP activity and calcium deposition in hMSCs, suggesting a suppressive effect on osteogenic differentiation [23].”

In Result, section authors have to rephrase the heading 2.2 scientifically. The interpretation of the results is quite complicated for the readers.

A. We have changed heading 2.2 in line 107 of the Results to read, “Effect of MDZ on temporal changes in gene expression of PPU-7 cell line”.

Figure-6: a, b SEM images are quite visible but c is not in good resolution. Authors have to replace it or omit.

A. We omitted Figure 6c according to your comment.

In conclusion, authors have to share some lines on future direction.

A. We have corrected and improved lines 560 to 565 of the Conclusion to read: “Further studies are required to elucidate the pharmacokinetic and pharmacological efficacy of MDZ in animal experiments. In the dental field, these findings support the repositioning of MDZ to promote dentin regeneration for endodontic treatments, such as the pulp capping. Moreover, these findings support advancing research from pig to human experimental models using human DPSCs to discover MDZ’s potential not only for future dental treatments, but also for organ regenerative medicine”.

Again, thank you for giving us the opportunity to strengthen our manuscript with your valuable comments and queries. We have worked hard to incorporate your feedback and hope that these revisions persuade you to accept our submission.

*English proofreading of this manuscript has been carried out by ACS ChemWorx Authoring Services.

Reviewer 2 Report

This study showed the potential for drug repositioning of MDZ for dentin regeneration at the cell biological, genetic, protein and crystal engineering levels. The research design is straightforward and easy to follow and the data are also solid. Overall, the manuscript is well written and the findings are very interesting and have some impacts on clinical dentistry. The following is my minor concern.

It is not clear how the authors are interested in MDZ as a possible candidate for dentin regeneration. For examples, MDZ could regulate the ion channel, which also affects the cellular differentiation. Authors should provide some comments on the possible roles of MDZ.

Author Response

The authors would like to thank the second reviewer for the constructive comment and helpful suggestion. Our response to the reviewer’s comment is in the attached file.

Response for the second reviewer

This study showed the potential for drug repositioning of MDZ for dentin regeneration at the cell biological, genetic, protein and crystal engineering levels. The research design is straightforward and easy to follow and the data are also solid. Overall, the manuscript is well written and the findings are very interesting and have some impacts on clinical dentistry.

The authors would like to thank the second reviewer for the constructive comment and helpful suggestion. Our response to the reviewer’s comment is as follows:

It is not clear how the authors are interested in MDZ as a possible candidate for dentin regeneration. For examples, MDZ could regulate the ion channel, which also affects the cellular differentiation. Authors should provide some comments on the possible roles of MDZ.

A. We have corrected the text in lines 332-336 of the Discussion to read: “In vitro study using human mesenchymal stem cells (hMSCs) has shown that MDZ inhibits ALP activity and calcium deposition in hMSCs, suggesting a suppressive effect on osteogenic differentiation [23]. However, little has been known about the effect of MDZ on the properties of dental pulp tissues. We therefore hypothesized that MDZ affects odontoblastic differentiation rather than the osteogenic differentiation of dental pulp stem cells.”

Reviewer 3 Report

Karakida et al. did a nice piece of work evaluating the prospects of Midazolam for dentin regeneration. 

The article is of good quality and presented data are plausible. I could see the highly polycrystalline 

structures and nanoparticles in the TEM and SAED preparations but that is not sufficient to determine 

overall quality of work. So, I have some concerns which can be helpful to authors to improve the overall 

quality of the work.

Major Concerns

1. What methods are used to determine the ALP activity in Figure 1B is it the ALP staining or 

the qPCR? Authors should clearly mention the experimental tool that generated the bar 

diagram. It is strongly recommended to provide the ALP staining of cell culture to enhance 

the quality of work. 

2. Bmp2 and TGFB1 were previously implicated in dentine regeneration. However, authors 

finding suggested that they are not so significant in combination with MDZ compared to 

MDZ alone. Did authors try to induce the ALP activity in the PPU-7 cell line using BMP2 and 

TGFb1 alone to rule out the bias in the experiment? What may have caused the reduced ALP 

activity in culture treated with combination of MDZ+BMP2 and MDZ+TGFb1? 

3. For possible use in dentine regeneration in-vivo, heterogeneous cell populations need to 

coordinate in consort to differentiate into the odontoblast like cells to form the reparative 

dentine. Did authors perform any co-culture experiment that also showed more or less 

similar results that would increase the possibility of MDZ being used for drug repositioning?

4. I strongly suggest the authors to perform pulp access cavity experiment with drug 

positioning and show at least gene expression changes that greatly increase the value of this 

work.

Minors:

1. If authors can provide high resolution images of SEM and TEM, it would increase the overall 

paper quality.

Author Response

The authors wish to thank the third Reviewer for their helpful and constructive suggestions, especially those to help clarify the presentation of our findings and the conclusions we have drawn. Our response to the reviewer’s specific comments is in the attached file.

Karakida et al. did a nice piece of work evaluating the prospects of Midazolam for dentin regeneration.

The article is of good quality and presented data are plausible. I could see the highly polycrystalline structures and nanoparticles in the TEM and SAED preparations but that is not sufficient to determine overall quality of work. So, I have some concerns which can be helpful to authors to improve the overall quality of the work.

The authors wish to thank the third Reviewer for their helpful and constructive suggestions, especially those to help clarify the presentation of our findings and the conclusions we have drawn. Our response to the reviewer’s specific comments is as follows:

 Major Concerns

 1. What methods are used to determine the ALP activity in Figure 1B is it the ALP staining or the qPCR? Authors should clearly mention the experimental tool that generated the bar diagram.

A. We used a quantitative colorimetric method with a p-nitrophenylphosphate as the substrate. We have corrected the text in lines 440 to 449 of the Materials and Methods to read: “We measured ALP activity in each well as described previously [43]. The cells were plated on a 96-well plate at a density of 3.16 × 104 cells/cm2 and were cultured in the standard medium for 24 hrs. The medium was changed to growth medium supplemented with 0, 2.5, 5, or 10 μM of MDZ with 500 ng/mL of rhBMP2 (R&D Systems, Minneapolis, MN, USA) or with 1 ng/mL of rhTGF-β1 (Cell Signaling Technology, Danvers, MA, USA). After 72 additional hours of incubation, the cells were washed once with PBS, and ALP activity was assayed using 10 mM p-nitrophenylphosphate as the substrate in 100 mM 2-amino-2-methyl-1,3-propanediol-HCl buffer (pH 10.0) containing 5 mM MgCl2 and incubated for 10 minutes at 37°C. Adding 0.2 M NaOH quenched the reaction, and the absorbance at 405 nm was read on a plate reader.

 A. We have also corrected the text in lines 83 to 85 of the Results to read” “we investigated the effects of MDZ on ALP activity in the PPU-7 cell line by using a quantitative colorimetric method with a p-nitrophenylphosphate as the substrate.”

It is strongly recommended to provide the ALP staining of cell culture to enhance the quality of work.

A. We performed the ALP staining of cell culture and indicated its result in Fig. 1c. We have added the sentence to read in lines 95 to 98 of Results to read: “ALP staining for the mineral-induced PPU-7 cells displayed blue-colored staining images (Figure 1c). The cells were densely distributed on the plate of MDZ-only, whereas the combination of MDZ+BMP2 and MDZ+TGF-β1 displayed the low density.” 

A. We have newly created the heading in 451 of the Results to read, “4.2. Alkaline phosphatase staining” and added the sentence in 452 to 456 to read, “The cells were spread on a 6-well plate at density of 3.16 × 104 cells/cm2. After incubation for 24 hrs, the medium was changed to mineralization medium. After an additional 2 days of incubation, the cells were rinsed twice with PBS, fixed with 10% formaldehyde for 30 min, stained with 0.1 mg/mL of naphthol AS-MX phosphate, 0.5% N,N-dimethylformamide, fast blue BB salt, and 2 mM MgCl2 in 0.1 M Tris-HCl buffer (pH 8.5) for 30 min at room temperature, and then washed with dH2O and photographed.”

2. Bmp2 and TGFB1 were previously implicated in dentine regeneration. However, authors finding suggested that they are not so significant in combination with MDZ compared to MDZ alone. Did authors try to induce the ALP activity in the PPU-7 cell line using BMP2 and TGFb1 alone to rule out the bias in the experiment?

A. In order to avoid the confusion for readers:

(1) We added the experimental condition in Fig. 1.

(2) We indicated the ALP activity as increasing or decreasing ratios relative to the level of the control (i.e., 0 μM MDZ, 0 μM BMP2, 0 μM TGF-β1), which was set at 1.

(3) To compare the ALP activity in the PPU-7 cell line using BMP2 and TGF-β1 alone, we added a dotted line. Then, we have corrected the text in lines 87 to 95 of Results to read: “When the ALP activity level of the control (i.e., 0 μM MDZ, 0 ng/mL rhBMP2, 0 ng/mL rhTGF-β1) was set at 1.0, the addition of MDZ-only significantly enhanced ALP activity in PPU-7 cells in a concentration-dependent manner, especially the ALP activity at 10 μM MDZ (i.e., 10 μM MDZ, 0 ng/mL rhBMP2, 0 ng/mL rhTGF-β1), which was 1.75-fold higher than the control. BMP2-only (i.e. 0 μM MDZ, 500 ng/mL rhBMP2, 0 ng/mL rhTGF-β1) or TGF-β1-only (i.e. 0 μM MDZ, 0 ng/mL rhBMP2, 1 ng/mL rhTGF-β1) also slightly or significantly enhanced the ALP activity. The combination of rhBMP2 or rhTGF-β1 with MDZ (MDZ+BMP2 or MDZ+TGF-β1) (5 or 10 μM) slightly increased ALP activity (1.14-1.27-fold for MDZ+BMP2, and 1.28-1.29-fold for MDZ+TGF-β1) compared to control.”

(4) We slightly corrected the description in Fig. 1.

What may have caused the reduced ALP activity in culture treated with combination of MDZ+BMP2 and MDZ+TGFb1?

A. As we discussed (line 341 to 351 of Discussion), we have assumed that the reduction of ALP activity in culture treated with combination of MDZ+BMP2 and MDZ+TGF-β1 might be due to the susceptibility and/or reactivity for GABAA receptor, although we don’t know if it’s mechanism arises from the competitive action against the receptor and/or the structural changes of MDZ by binding to BMP2 or TGF-β1.

3. For possible use in dentine regeneration in-vivo, heterogeneous cell populations need to coordinate in consort to differentiate into the odontoblast like cells to form the reparative dentine. Did authors perform any co-culture experiment that also showed more or less similar results that would increase the possibility of MDZ being used for drug repositioning?

A. Thank you very much for your valuable suggestion. Unfortunately, we have not established any systems for a co-culture experiment. In the present study, we used porcine dental pulp cells to gain the basic information of MDZ for drug repositioning. Our future challenge is to discover the potential for drug repositioning of MDZ with human dental pulp stem cells (DPSCs). We wish to establish the system for co-culture experiment at that time.

 4. I strongly suggest the authors to perform pulp access cavity experiment with drug repositioning and show at least gene expression changes that greatly increase the value of this work.

A. The authors have also realized that the reviewer’s comment is a great and fruitful suggestion. However, the authors wish to point out the above study mostly address the basic experiment in the present study. We have corrected the text in lines 560 to 565 of Conclusions to read: “Further studies are required to elucidate the pharmacokinetic and pharmacological efficacy of MDZ in animal experiments. In the dental field, these findings support the repositioning of MDZ to promote dentin regeneration for endodontic treatments, such as the pulp capping. Moreover, these findings support advancing research from pig to human experimental models using human DPSCs to discover MDZ’s potential not only for future dental treatments, but also for organ regenerative medicine”.

Minors:

1.      If authors can provide high resolution images of SEM and TEM, it would increase the overall paper quality.

A. We omitted one of SEM images, Figure 6c, according to the request from other reviewer since it was low resolution. Each nanoparticle in the precipitates was evidently observed in Figure 6b, therefore, this omission has no effect on conclusion.

For TEM images of Figure 8b and 9b, we digitally zoomed the images to some extent captured in lower magnifications. This is because the observation under quite high magnification resulted electron-bombardment damages for crystals within short time (less than 30 s).

Again, thank you for giving us the opportunity to strengthen our manuscript with your valuable comments and queries. We have worked hard to incorporate your feedback and hope that these revisions persuade you to accept our submission.

Round  2

Reviewer 1 Report

Well improved. Just check minor English correction. 

Reviewer 3 Report

All the contents were well prepared and modified for publications in IJMS.

I strongly suggest this manuscript to be published as it is.